# Scalable Knowledge Management to Meet Global 21st Century Challenges in Agriculture

Nicholas M. Short, Jr. [1], M. Jennifer Woodward-Greene [2] , Michael D. Buser [3,4] and Daniel P. Roberts [4,5,*]

1   National Government Unit, Environmental Systems Research Institute, Beltsville, CA 92373, USA
2   National Agricultural Library, United States Department of Agriculture, Agricultural Research Service, Beltsville, MD 20705, USA
3   Office of National Programs, United States Department of Agriculture, Agricultural Research Service, Beltsville, MD 20705, USA
4   Partnerships for Data Innovations Initiative, United States Department of Agriculture, Agricultural Research Service, Beltsville, MD 20705, USA
5   Sustainable Agricultural Systems Laboratory, United States Department of Agriculture, Agricultural Research Service, Beltsville, MD 20705, USA
*   Correspondence: dan.roberts@usda.gov; Tel.: +1-301-504-5680

**Abstract:** Achieving global food security requires better use of natural, genetic, and importantly, human resources—knowledge. Technology must be created, and existing and new technology and knowledge deployed, and adopted by farmers and others engaged in agriculture. This requires collaboration amongst many professional communities world-wide including farmers, agribusinesses, policymakers, and multi-disciplinary scientific groups. Each community having its own knowledge-associated terminology, techniques, and types of data, collectively forms a barrier to collaboration. Knowledge management (KM) approaches are being implemented to capture knowledge from all communities and make it interoperable and accessible as a "group memory" to create a multi-professional, multidisciplinary knowledge economy. As an example, we present KM efforts at the US Department of Agriculture. Information and Communications Technology (ICT) is being developed to capture tacit and explicit knowledge assets including Big Data and transform it into curated knowledge products available, with permissions, to the agricultural community. Communities of Practice (CoP) of scientists, farmers, and others are being developed at USDA and elsewhere to foster knowledge exchange. Marrying CoPs to ICT-leveraged aspects of KM will speed development and adoption of needed agricultural solutions. Ultimately needed is a network of KM networks so that knowledge stored anywhere can be used globally in real time.

**Keywords:** agriculture; communities of practice; food security; geographic information systems (GIS); information and communication technology (ICT); knowledge graphs; knowledge management

## 1. Introduction

Sustainably achieving global food security is a daunting challenge [1–6]. The task of supplying adequate calories is not being met at present with approximately 800 million undernourished people globally and half the world population lacking access to one or more essential nutrients [3,7]. Additionally, current agricultural systems are not sustainable because they are resource intensive and a leading cause for environmental transgressions [6,8–10]. Globally, agricultural systems are a primary driver of climate change, land-use change, biodiversity loss, depletion of fresh water, and pollution of land and waterways through nitrogen, phosphorus, manure, and pesticide run-off from agricultural fields [9,11–13]. Sustainably achieving global food security will only get more challenging with a population expected to increase by one to two billion people by 2050. Global income is also expected to increase three- to four-fold and will result in diet diversification to include more meat and the associated need to produce more plant-based calories [3,4,14].

Agricultural solutions are needed that increase production of nutritious food globally and decrease the impact of agricultural production systems on the environment, both despite an increasingly erratic climate [15–19]. Food production doubled over the past several decades largely due to the use of irrigation, inorganic nutrients to manage soil fertility, synthetic chemical pesticides to control pathogens and pests, mechanical loosening of soil, and development of high-yielding crop cultivars [20–23]. New approaches and technologies are needed given the negative impacts of these production systems on the environment and, in some regions, a decline in crop yields [6,9,21,24–26]. There needs to be a focus on ecological intensification and other approaches/technologies directed at minimizing anthropogenic inputs and protecting and improving soil and water resources, as well as on agricultural intensification and climate resilience [1,9,12,15,20,27]. There is also a need to expand crop cultivar development programs so that yield and nutritional quality are maximized, together with resistance to biotic and abiotic stresses, and efficient water and soil nutrient use [18,28–30]. The high-yielding cultivars currently in use contain calorie-rich macronutrients such as fat, protein, and carbohydrate but not necessarily adequate micronutrients (vitamins, minerals, essential amino acids and fatty acids) and other health-benefiting phytochemicals (phenolics, alkaloids, organosulfur compounds, phytosterols, carotenoids) that are needed to meet human nutritional needs and decrease chronic diseases such as heart disease, stroke, diabetes, Alzheimer's, cataracts, and age-related functional decline [31–33].

## 2. Knowledge Management for Development and Adoption of Agricultural Solutions

For global food security, a better use of natural (soil, water, air, genetic diversity) and, importantly, human resources (knowledge) are needed [34–38]. Knowledge that has emphasis on ecological and agricultural technology innovations needs to be created, and existing and new knowledge deployed, and adopted by farmers and other decision makers in the global agricultural enterprise. Creation of new knowledge will require collaboration amongst many professional communities world-wide including farmers and other agriculturalists, agribusinesses, economists, policymakers, scientists, and health professionals [27,38,39]. Within the scientific community, scientists from multiple disciplines will need to collaborate on plant breeding or other genetic approaches for cultivar development, on developing new crop management systems, weed and pest control strategies, and more efficient water use approaches to name a few [1,5,12,27].

Professional communities and scientific disciplines each have their own knowledge with associated terminology, techniques, and forms of data and models which form barriers to multidisciplinary efforts [39]. To accelerate progress towards agricultural solutions there is a need to integrate and manage knowledge from these communities to develop a new effective multi-professional, multidisciplinary knowledge economy [37,39,40]. The scientist, farmer, and other workers employed in the agricultural enterprise are, after all, knowledge workers who apply relevant information gleaned from many professional communities. Explicit knowledge, knowledge that is written or codified, and tacit knowledge, knowledge that is experiential or otherwise non-written or non-codified, must be captured and made interoperable and accessible as a group memory of all facets of this multi-professional, multidisciplinary knowledge economy through Knowledge Management (KM) approaches. In this way, "KM can be seen as an effort to create an information environment . . . that has rich, deep, and open communication and information access . . . The logical conclusion is to attempt to apply these same successful environmental aspects to knowledge workers at large, and that is precisely what KM attempts to do" [41]. Knowledge management is increasingly recognized to be important in agriculture and other portions of the world development sector since knowledge-intensive enterprises that manage knowledge can be more efficient and have increased innovation potential; agriculture being information and knowledge intensive [37,42].

Knowledge management promotes an integrated approach to identifying, capturing, evaluating, retrieving, and sharing all of an organization's, enterprise's, and whatever

else is relevant knowledge and information assets. These assets may include databases, documents, policies, procedures, or previously un-captured expertise and experience of individual workers [43]. Knowledge management can be broken down into the following operational components: (1) content management, or management of information assets; (2) expertise location, or systems to find experts and desired knowledge; (3) capturing tacit knowledge embedded in personal expertise and making it explicit to ensure what is learned is passed on to others; and (4) communities of practice (CoP), groups of people with shared interests that come together virtually or in person to discuss problems and opportunities, best practices, and lessons learned [41,44,45].

Information and Communication Technology (ICT) is particularly useful in KM for agriculture by facilitating sharing knowledge and sources of knowledge among geographically dispersed people or units of organizations such as farmers, scientists, policy makers, and other constituents of the agricultural enterprise [37,41,42,46]. The ICT-leveraged KM approaches should help reduce the fractured state of the agricultural enterprise where information and knowledge currently exist in siloes. The ICT provides a resource for centralized storage repositories and management of content while data mining techniques allow extracting knowledge associated with different members of the agricultural community or identification of experts [42]. Gruber [47] suggested that ICT allows collaboration where workers "search for, collect, organize, share and collaborate around information" leaving information behind in a group memory that gets managed by appropriate CoPs—as tacit knowledge gets used, reused and curated, it transforms from being tacit to explicit knowledge. For Gruber, the key to collaboration leveraged by ICT is to approach the field of Artificial Intelligence (AI) differently; the field should focus on Humanistic AI to augment human intelligence and the acquisition of knowledge.

Nonetheless, challenges are present with ICT-leveraged KM approaches such as with ICT inaccessibility among certain members of the agricultural community and problems with data heterogeneity, inconsistency, completeness, and privacy concerns [37,48]. Moreover, KM initiatives need to be implemented that encourage iterative and inclusive communication between the communities of experts, such as scientists, and farmers and other decision makers to effectively transfer knowledge that is seen as relevant and credible by all participants. This ensures equity and democratization of the knowledge processes [37,40,42]. Effectiveness decreases when communication is largely one-way between expert and decision maker communities. The ability to transfer knowledge also decreases when communication is infrequent or occurs only at the outset and when members of expert or decision maker communities feel excluded or marginalized [37,40].

Irrespective of the involvement of ICT, without iterative and inclusive communication between these communities to capture explicit and tacit knowledge, experts may incorrectly assume what solutions are needed, may work on solutions that are no longer relevant, and resultant excluded parties may question the legitimacy of information that is generated [40]; this would lead to the need for more formal knowledge acquisition methodologies to ensure proper capture of explicit and tacit knowledge. Mulder et al. [49] suggest a methodology for transforming tacit knowledge from a community of experts into explicit knowledge by recognizing the need to provide context through the alignment with organization goals (i.e., a bounded environment), the production of a controlled vocabulary, iterative development amongst the experts, and a teleological purpose, all implemented in a common database or group memory.

It follows that KM approaches in agriculture need to be scaled by leveraging ICT and, importantly, other methodologies to allow multilateral knowledge and information flow globally among research institutions, other institutions involved in the agricultural enterprise, and the knowledge bases of the communities of farmers and other decision makers. Because of cost constraints in agriculture, automated ICT-based KM approaches that include AI, Machine Learning (ML), Deep Learning, blockchain, and Internet of Things (IoT) will be needed; evolving agriculture into what some are calling "Agriculture 5.0" [50].

**3. Knowledge Management in Business and Academia—Takeaways for Integration of Knowledge Management in Agriculture**

Knowledge management as a discipline emerged from two different approaches that map well to both the business and science aspects of agriculture. These approaches balanced the need for cost optimization of managing knowledge in business and the need for precision in science. The business approach focused on unstructured knowledge (i.e., unstructured data) while the scientific side tended towards more structured approaches, where knowledge is carefully and methodically curated.

*3.1. Emergence of KM in Business*

Knowledge management emerged in the 1990s out of the business consulting communities and companies as they attempted to support multi-national corporations challenged by operating in a variety of cultures, laws and regulations, geographies, and practices. Peter Drucker, who is considered the "father of modern business management", avoided definitional obfuscation by simply articulating that knowledge is information that is used by a knowledge worker, working in a system with corporate governance around quality and productivity [51]. As a result of Drucker's vision, business solutions involved the design and implementation of organizational governance, people and their skill sets and roles, information systems, and business processes to drive business outcomes in what IT practitioners of today call Enterprise Architecture.

Business firms very quickly realized the dependency on geographic solutions that helped manage the flow of data and information across national and cultural boundaries. That is, they modelled knowledge networks that span geographic boundaries in order to increase organizational performance [52]. When information systems were being designed, these companies and consulting firms also realized the need to simplify the formalisms from AI/computer science, mathematics, and philosophy academic disciplines, emphasizing the need to leverage fields like psychology and sociology to improve human-computer interfaces, resulting in user-centered design and social networking techniques that are commonplace today. To these firms, the key to scaling KM globally depends on the ability of the organization to simplify communication to promote collaboration across cultures and continents, while avoiding academic formalisms.

In the ICT industry, products such as IBM's Lotus Notes focused on simplicity and communication around early KM solutions that attempted to integrate human-centered, point solutions like document management, search and subscription, email, etc. Novel start-up companies like Intraspect emerged in the mid-1990s utilizing computer-human interface techniques for ease of knowledge capture, graph structures for organization, text mining for search and discovery, and web frameworks for dissemination of knowledge. That version of the business market converged onto Microsoft's Sharepoint technology. Overall, the global KM market is large; for example, the KM software market is expected to grow by $4.82B between 2021 and 2026 (KM Software Market Size).

*3.2. Evolution of KM Theory in Academia*

Unlike the engineering efforts modelled after the less formal approaches from the business consulting firms in KM, much of the academic KM effort focused on automation, tracing its roots to formalisms from cognitive science, computer science, and mathematics, which played a key role in AI. While KM may have been popularized by the business community, who were interested in productivity from these formalisms, its formal or theoretical side borrowed ideas from academia ranging from graph theory, formal or first order predicate logic, and derivative logics (e.g., temporal, modal, probabilistic, fuzzy, etc.), to computational theory.

Academic efforts in the ICT arena have focused on applying "how" and "why" questions to information (i.e., structure, context) about data (i.e., symbols) taking the form of if-then rules (i.e., called production rules), propositions or predicates, (i.e., logical representation) entities and relationships or associative networks (i.e., semantic networks),

heuristics for solving combinatoric problems, etc.; all falling under the AI subdiscipline of Knowledge Representation (KR). Starting with Semantic Networks such as KR, there has been an explosion of techniques like Conceptual Dependency Graphs (Schank 75), Connectionism [53–55], Ontologies [56,57], and knowledge graphs to name a few. Many of these techniques found their way into the expert system market in the 1980s and 1990s, where there were many attempts and methods of gathering domain-specific expertise into production rules. Most of these techniques generated collections of concepts and relationships, etc. called information spaces, and were used to measure the significance, impact, or scope of any given knowledge base (e.g., number of production rules, concepts, or relations in an information space, etc.).

It is widely accepted that the difficulty and labor-intensive nature of codifying knowledge into information spaces led to disappointing experiences in cost effectively developing business applications of expert system technology and ultimately, the collapse of the AI software market in that era. This was partly due to expert systems having been good at problem solving within narrow domains, but not effective when it came to simple, common-sense problems. Many researchers have argued the need to invest heavily in developing a "common sense knowledge base" where domain specific expert systems could leverage to improve problem solving. The best-known response was the Cyc project (https://en.wikipedia.org/wiki/Cyc, accessed on 1 February 2023), whose approach was to "assemble a comprehensive ontology and knowledge base that spans the basic concepts and rules about how the world works". Lessons from Cyc influenced the emergence of KR standards such as Resource Description Framework (RDF), OWL (Web Ontology Language), W3C's Simple Knowledge Organization System (SKOS) for the semantic web, SKOS-XL for linking lexical entities, etc., which made knowledge bases scalable if not remaining labor intensive for knowledge capture.

In the 1990s, simplification of protocols like http and content types like html led to the ability for almost anyone to develop knowledge content that could be easily shared. However, the ability to "search" across global boundaries and information spaces for content is what led to extreme growth and development of many information retrieval techniques [58] and the ultimate success of the internet. Through better search engine optimization techniques like result ranking, Google, who eventually focused on intelligent search retrieval, came to dominate the market [58]. Google's information space manages over 500 billion facts on 5 billion entities [59].

Perhaps the best example of the culmination of these techniques around ontologies, search or information retrieval across unstructured repositories, and simple query languages using natural language was the development of IBM's Watson, which culminated in its win on the Jeopardy show in 2011, beating two Jeopardy champions. Watson has since been applied to several domain-specific areas like medicine, tax preparation, and weather forecasting.

### 3.3. Scalable Knowledge Management through ICT

While the World Wide Web can be seen as a first-generation approach to scaling KM in support of global challenges, recent advances in cloud computing, Big Data technologies like Hadoop, high performance analytics, and AI/ML will usher in the next wave of KM technologies that could finally realize Peter Drucker's vision. Two specific technologies, that when combined, should have significant impact on KM in agriculture, namely Knowledge Graphs (KGs) as a form of KR and Location Intelligence as a form of Geographic Information Systems (GIS).

Knowledge graphs, also known as semantic networks, represent a network of entities (objects, events, situations, concepts, etc.) and their relationships. They trace their root to the Database Management Systems (DBMS) community through the introduction by Bachman of the network database model in the 1969 Conference on Data Systems Languages (CODASYL). While there were competing models around Hierarchical DBMS and Relational DBMSs (RDBMS) to network models, RDBMSs ultimately prevailed in the mar-

ket because of their ability for high transaction rates, simple query languages (e.g., SQL), and strong financial support from the banking and finance industry. In other words, the marketplace determined the winner over perhaps technically superior solutions from an academic perspective. As a result, commercially, KG approaches were initially relegated to niche markets that primarily served scientific communities through object-oriented models such as SmallTalk and its derivative DBMS's like ObjectStore. More recently, the need for scalability to meet query rates and data volume and variety at scale have led to the development of KG datastores that can be accessed via structured queries across information spaces.

Like KM, the emergence of GIS resulted from the need to manage geographic information, but as spatial maps tied to decision support systems. While not traditionally considered part of KM, early work in knowledge-based GIS can be traced back to the US Geological Survey (USGS) and the University of California, Santa Barbara KBGIS; which relied on expert systems, a spatial object store, and traditional data management approaches [60]. In the DBMS realm, extensions to query languages defined spatial operators such as geofencing queries (i.e., spatially delimited), which evolved into geo-enrichment services. In the KM space, query languages such as SPARQL for Resource Definition Formats (RDF), for example, have been extended to languages like GeoSPARQL for geofenced queries against RDF graph structures. When coupled with mobile applications, GIS is moving towards providing individuals with "situational awareness" or Location Intelligence, which is defined as "the process of deriving meaningful insight from geospatial data relationships to solve a particular problem" (https://en.wikipedia.org/wiki/Location_intelligence, accessed on 1 February 2023). In other words, deriving knowledge needed for a solution from geospatial relationships.

Similar to SPARQL, the open source OpenCypher query language, which is used to return paths from a labelled property graph as result sets, has been extended with spatial operators for both the open source OpenCypher and the supporting company, Neo4j. While considered as having deficiencies in the traditional GIS capabilities realm, these extensions form a basis for addressing modern, scalable, geographically driven KM systems for driving knowledge flows across international geographic boundaries.

Recently, Esri, the market leader in GIS, announced its ArcGIS Knowledge Server product, which combines KGs with the full power of an industrial grade, proven, scalable GIS. Built on an ArangoDB data store, and interoperable with Neo4j databases, ArcGIS Knowledge can leverage many of the non-geospatial techniques from the OpenCypher community ranging from food supply chains (e.g., Farm to Fork), disease tracking and early detection, to basic search and discovery. Combined with its extensive functionality, ability to manage a variety of geospatial asset types, global LivingAtlas data store, and large world-wide user community, the ArcGIS platform with knowledge server is a foundational technology for establishing true location intelligence in agriculture.

### 3.4. GIS-Based Knowledge Graphs

Although industrial-grade Geographic Knowledge Graph technology is relatively new, Janowicz et al. [61] presented their KnowWhereGraph framework as a tool for providing situational awareness to decision makers in disaster management scenarios, for providing immediate access to data about "food safety, wildfires, air pollution, worker health, supply chain disruptions, and transportation networks" for the agriculture sector, and for environmental intelligence. KnowWhereGraph relies on Esri's GeoEnrichment service, a GeoSPARQL query toolbox, and a Knowledge Graph-based geo-enrichment toolbox in ArcGIS Pro to access its rapidly growing data silos. The goal of KnowWhereGraph is to use KGs for linkages across multiple domains and use cases that require spatial data question and answering, but with ArcGIS Knowledge as the foundational product.

### 3.4.1. GIS-Based Knowledge Graph Information Search and Discovery Example

As proven by Google and others, KGs are particularly good at guiding users through query formulation for easy access of knowledge and are commonly used with semantic search techniques like autocompletion [62]. For example, managing/finding academic citations is inherently organized in a graph structure coupled with semantic search because of the reliance on nomenclature spanning a variety of disciplines. Given the spatial nature of agriculture, adding geospatial query capabilities via the power of GIS should increase query precision significantly. As an example of adding the geographic dimension, KM requirements around "finding an expert or information" can be implemented to facilitate collaboration as well as gathering information about a certain location. This is particularly important in agriculture where experts and scientists visit farmers on their farms for the exchange of knowledge (i.e., extension services, on-farm research). Information regarding the farm, or similar farms (size, soil type, crops grown, etc.) can be obtained prior to or during the visit to aid in knowledge exchange.

See Figure 1 as an example from the USGS of searching through an archive of publications and documents using KGs organized around citations. The figure depicts a citation network KG of publications using Esri's ArcGIS Pro as the GIS, thereby allowing users to query by location for experts who have a particular expertise as evidenced through their publications.

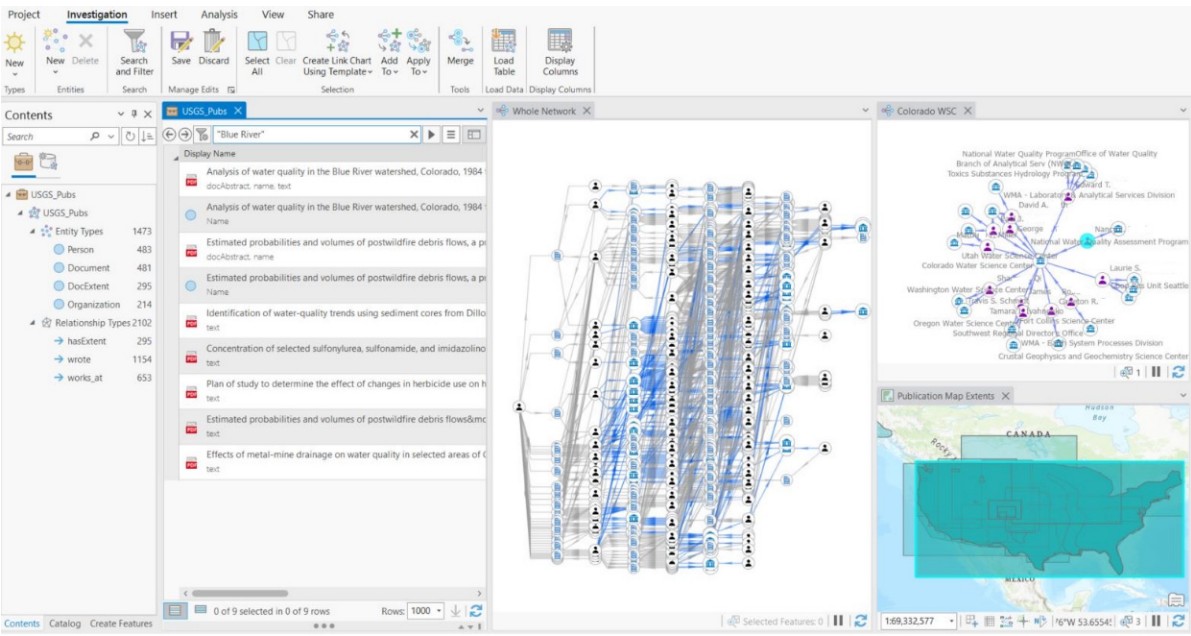

**Figure 1.** Determining where expertise is through a citation search at the US Geological Survey.

### 3.4.2. An Example Knowledge Product: Disease Surveillance

Although a knowledge product has been defined as "tools that facilitate the use of model outputs" [34], we use a more object-oriented definition. We define a knowledge product as a chunk or object that encapsulates a particular knowledge workers activity (e.g., a project) and includes their data, information, and knowledge assets. As an example, Figure 2 represents a knowledge product generated by a food safety regulator who traced the outbreak of Cyclosporiasis in leafy greens, examining the food supply chain from farm to table. In this case, the knowledge product consists of data captured from mobile devices and field investigations, restaurant and farm locations, vendors, and supply chain links from suppliers to vendors.

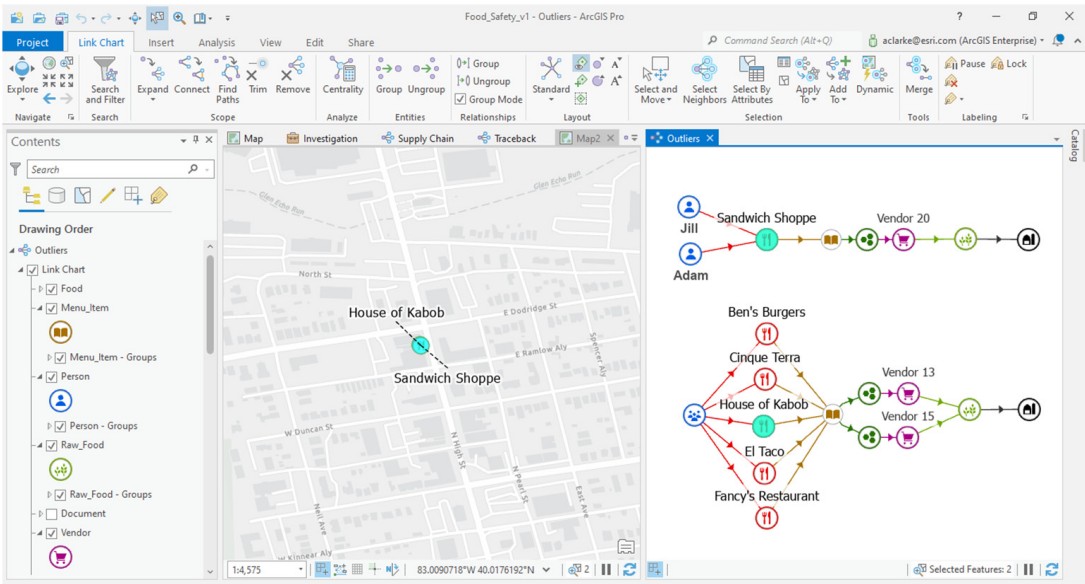

**Figure 2.** Using GIS and KGs to trace disease pathways for an outbreak of Cyclosporiasis.

The value of combining GIS with KGs is illustrated in Figure 2, whereby querying graph structures, scientists or analysts can infer spatial relationships between graphs that appear different from each other. In other words, perform social network analysis using common graph theory techniques to identify super spreader nodes (e.g., restaurants) that leverage graph centrality, or namely, those nodes that are important. Using betweenness centrality to quantify "the number of times a node acts as a bridge along the shortest path between two other nodes" (https://en.wikipedia.org/wiki/Centrality, accessed on 1 February 2023), an analyst can discover subgraphs that trace disease pathways from farm to restaurant. When the geospatial dimension is included, two apparently disconnected paths can be connected due to some of the nodes being located at or near each other. In Figure 2, for example, the location of the "Sandwich Shop" and the "House of Kabob" each had outbreaks but are only connected because they were in the same location (left side of Figure 2). From a practical perspective, isolating the outbreak to one location reduces investigation time, leading to a quicker response to the outbreak.

The results of this analysis would then be encapsulated into a specific object or knowledge product and checked back in the knowledge base for reuse by other regulators during the next outbreak. Capture of tacit knowledge into knowledge products would then follow the multi-level framework suggested by Mezghani et al. [63], with particular emphasis on a knowledge engineer helping capture metadata about the expert(s), to facilitate continued, interactive acquisition across domains [64].

## 4. Knowledge Management in Agriculture

Various organizations in the agricultural enterprise have been considering KM and advances in ICT around KM. Fortunately, lessons learned in business and ICT technology developed for KM are applicable to agriculture.

### 4.1. Agricultural Thesauri

Bridging different cultures and facilitating international knowledge flows is essential for facilitating interoperability across location-specific knowledge repositories, and a basis for information search and discovery across multi-lingual boundaries. Thesaurus-driven and other efforts in agriculture employing linked data and semantic web standards have proven invaluable for data and knowledge interoperability and discovery especially across cultures and continents [65].

Early KM initiatives in agriculture include the three major agricultural thesauri. (1) AGROVOC, from the Food and Agriculture Organization (FAO) of the United Nations, which was started in the 1980s to facilitate international knowledge flows. This initiative started in the form of print catalogues for describing documents, etc., and evolved into a multi-lingual thesaurus and controlled vocabulary consisting of over 40,000 concepts and 900,000 terms in 41 languages (https://www.fao.org/agrovoc/about, accessed on 1 February 2023). (2) The CAB Thesaurus (https://www.cabi.org/cabthesaurus, accessed on 1 February 2023) in the UK, which has been in use since 1983 and currently contains over three million English terms. And (3), the National Agricultural Thesaurus (https://agclass.nal.usda.gov, accessed on 1 February 2023) from the USDA, National Agricultural Library (NAL), which is of the same era. The first digital National Agricultural Thesaurus version appeared in 2002 in English and has been available in Spanish since 2008. All three of these agricultural thesauri have historically been used for subject indexing at each of their home institutions, for total coverage estimated at over 25 million bibliographic records. This is in addition to other organizations using the terms for their subject indexing [66].

The FAO, CABI, and NAL have a long history of collaboration and innovation to enhance semantic web interoperability. The concept data in the three thesauri use persistent uniform resource identifiers (URIs) as a single label representing each concept in all its forms (languages, synonyms, related terms, etc.) and these URI are linked by extensive mappings of mutual concepts between all three thesauri. Together the FAO, CABI and NAL curators developed the Global Agricultural Concept Space or 'GACS' as a namespace of concepts relevant to food and agriculture, which included the creation of GACS first and only subscheme, GACS Core (http://browser.agrisemantics.org/gacs/en/, accessed on 1 February 2023). The fundamental idea behind selection of terms in GACS is essentially a Venn diagram consisting of the most frequently used (i.e., important) concepts in agriculture from these three resources, based on the subject indexing of the millions of records managed by FAO, CABI, and NAL. Curation of GACS ceased in 2016, but the vision for GACS lives on in National Agricultural Library Thesaurus (NALT), which was first published as the NALT Concept Space (NALT) in 2022, with its first sub-scheme, NALT Core, based on GACS.

With the advent of graph technology, whether as RDFlabelled property graph databases such as Neo4j—or Wikidata type graphs, the data sources include more and more entities, and as many relationships between them as possible, for an incredible, complex web of knowledge. The ability to add information, including properties that are optional, i.e., not constrained by rows and columns order, can be challenging for data consumers. Shape Expressions (ShEx) allows applications and users to declare what should be in the RDF—and validate against that standard [67].

### 4.2. Embedding Knowledge Management in Ag Research Institutions: USDA

The Agricultural Research Service (ARS) is USDA's main in-house research agency, and one of the largest agricultural research organizations in the world. Like the multi-national corporations of the 1990s, USDA-ARS faces geographic and cultural challenges regarding agency data, information, and knowledge use. This is due to the wide-ranging locations of its research centers within the United States, and throughout the world, and the multi-disciplinary scientific research efforts conducted by its scientists and collaborators; with each scientific discipline having its own culture (own knowledge with associated terminology, techniques, and forms of data and models).

Partnerships for Data Innovations (PDI) was formed out of recognition that USDA-ARS needed to scale through better data management to meet the demands of the agricultural community. The ARS scientists and collaborators produce large volumes of Big Data (data that varies in volume, variety, velocity) from many sources (institutional center-based and farm-based) that is siloed, geographically dispersed, and unmanaged. The PDI is a USDA-ARS enterprise-wide research architecture initiative, and associated staff, implemented to efficiently leverage geographically dispersed ARS- and collaborator- multidisciplinary

research operations and accelerate agricultural research through standardization, automation, and integration of this data [68]. It is also designed to balance the need for expediting on-farm research while addressing the concerns around de-identifying farm geospatial data to protect privacy and individual producer competitive advantage [69]. Finally, PDI is by design a partnership between government, academia, and the agricultural business community attempting to ensure information and knowledge flows between these aspects of the agricultural enterprise. In other words, PDI is creating a test bed for the KM concepts presented in this paper.

While PDI emerged out of the need to capture and curate unmanaged, highly valuable Big Data from scientists, it is morphing into a KM initiative providing easy-to-use tools for the next generation of scientists and collaborators to capture data, information, and knowledge as a byproduct of their daily work. The KM approaches are needed because (1) as mentioned above, agency knowledge and information assets are currently siloed within locations and scientific disciplines, (2) the task of finding desired experts and desired knowledge and information needed by scientists, policymakers, and farmers and other end-users can be a challenge, and (3) tacit knowledge possessed by agency staff and collaborators is not being captured and is at risk of being lost due to retirements, etc. Relying on the previously discussed approaches from the business world, the formalisms from academia, and the ICT technology developed for KM, the KM at the ARS is effectively becoming the process of capturing tacit and explicit knowledge from scientists and collaborators and transforming it into managed explicit agricultural knowledge products for eventual curation at the NAL where it will be available for use, with permissions, by members of the agricultural enterprise.

### 4.2.1. Decision Support Informatics (DSI) Platform for Knowledge Management

To support this process of capturing tacit and explicit knowledge and transforming it into curated knowledge products at NAL, the PDI is developing a modern, industry-standard, geo-spatial cloud-based system, which integrates diverse database networks and facilitates data sharing and cooperation among participating researchers into a geospatial framework. This Decision Support Informatics platform (DSI) increases efficiency and cost savings by bringing siloed research into an enterprise research platform built on USDA's standard around GIS and the Federal Office of the Chief Information Officer (OCIO) emphasis on being cloud-first. This infrastructure also provides an environment or field-oriented platform for other technology organizations to build innovative applications, including IoT sensor data for field and other Big Data collection to help scientists and farmers easily capture data; mobile data capture for disease management; satellite and drone imagery for measuring the impact of soil regenerative practices, and machine data capture for crop health and yield estimation, to name a few. Other features will include Find and Ask the Expert which allows ARS scientists and collaborators to locate experts with tacit and explicit knowledge. The resulting cloud-based, logical architecture (Figure 3), after pipelines are developed between DSI and other Agency IT infrastructure assets, directly reflects the described knowledge pyramid [70]. Data and data feeds are captured and stored in the Agriculture and Food System layer; searchable, geospatial information fed from the lower layer reside in the Data Hub Middle Layer; and knowledge is consumed in the DSI layer. As with any data management platform, The Data Hub Middle Layer is designed to drive data standards, data interoperability, security, and cost efficiency. Furthermore, also shown is the AI and Analytics Tools Sandbox that is used to produce computationally intensive information products like geospatial analytic models suggested in Fitzgibbon et al. [71]. In other words, the architecture models the basic workflow of KM for capturing, organizing, sharing, and reusing knowledge derived from data and models, all within a corporate governance framework.

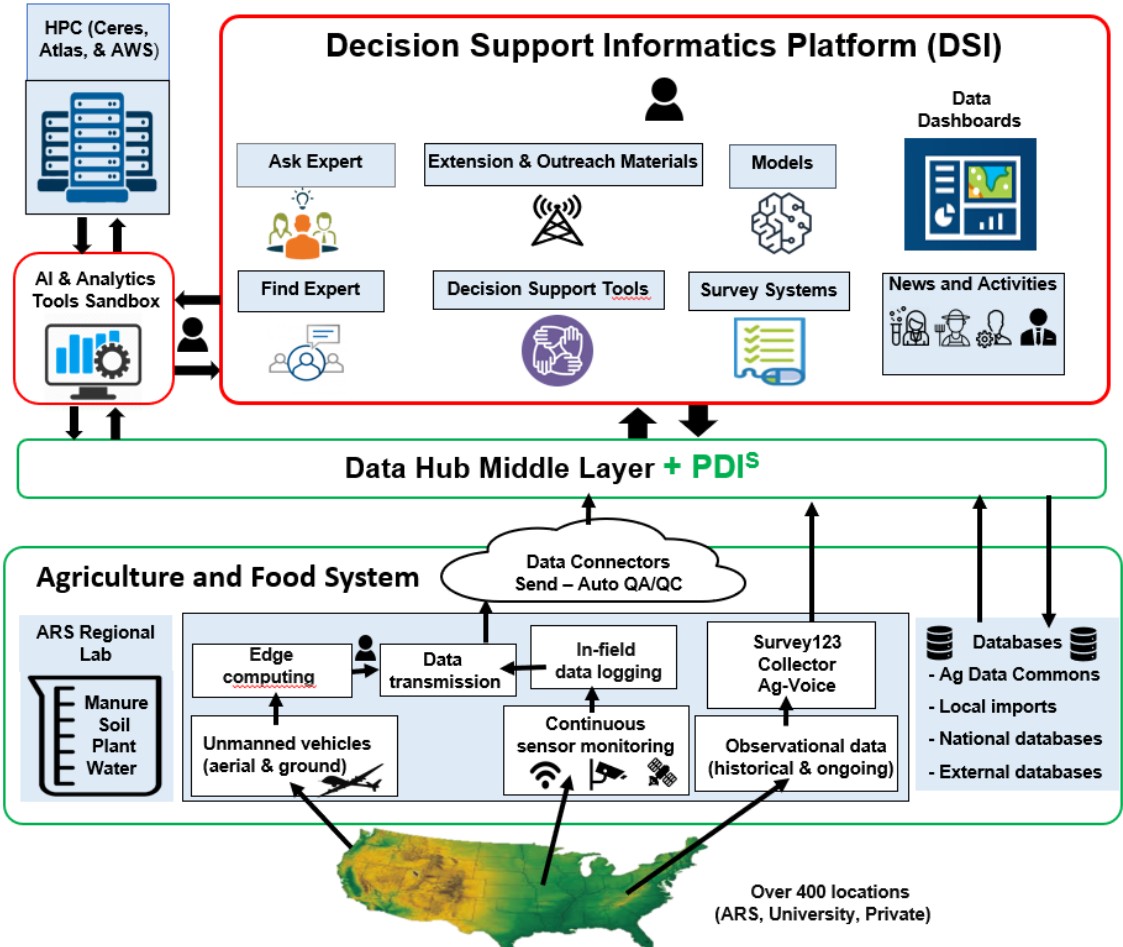

**Figure 3.** The Decision Support Informatics platform approach to Knowledge Management.

As a result of the DSI platform approach to IT and KM, numerous applications have already been delivered that should reduce the time it takes to deliver knowledge and information assets from and to ARS and collaborator field laboratories.

### 4.2.2. Examples of Capturing Data and Other Knowledge Assets

Partnerships for Data Innovations developed a data ingestion tool, the Farm App, that allows rapid entry and capture of georeferenced farm operations data (e.g., tillage operations, fertilizer regimens, irrigation regimens). By working with personnel at six ARS farm research locations, the PDI staff integrated functionalities into the Farm App that meet a wide range of needs while keeping the app customizable for integration of additional location-specific requirements if necessary. Farm operations personnel can now easily enter information regarding each field worked via a form-based mobile application that automatically captures location information for real-time or offline uploading into the DSI platform for long-term curation. As the Farm App is rolled-out and used on ARS locations nationally, the agency will have an increasingly comprehensive interoperable knowledge store regarding fields and field operations for use by scientists and other personnel during day-to-day tasks.

Other examples of how the PDI is using grass-roots efforts for development of data and information solutions to standardize and make data interoperable, as well as store information and knowledge for Agency-wide and collaborator access, are the PDI Dirt to Shirt and USDA Cattle Fever Tick Eradication programs [68]. The U.S. cotton industry is collaborating with the PDI on the Dirt to Shirt program to develop field data collection apps for fiber quality, agronomic practices, soil, and weather data to create an interoperable

data and knowledge store on the DSI platform that used to be housed at disparate industry and USDA-ARS locations. Data query tools, analytics, and output dashboards will make all data and knowledge easily accessible to team members. Regarding the Cattle Fever Tick Eradication program, developed between the PDI, USDA—ARS, and USDA—APHIS, APHIS inspectors can nearly instantaneously determine the geographic extent of cattle fever tick outbreaks in the United States and implement proposed quarantine maps for disease management due to the development of web mapping software applications by the PDI in collaboration with industry partners. Historically, this process would take six or more weeks to complete [68].

To capture agency tacit knowledge, the PDI is rolling-out Protocols.Io in collaboration with NAL. Protocols.Io offers each ARS scientist a dedicated workspace to create, maintain, organize, and share methods and Standard Operating Procedures (SOPs) across various research projects and organizational units. In addition to aiding in efficiency of collaboration, Protocols.Io allows capture of tacit knowledge of specific reagents to use, exact protocol methods, etc., that are not detailed in scientific publications or via other information media.

### 4.2.3. Organizing Data and Information Assets

As geography is the organizing principle or method for contextualizing incoming data from the aforementioned products, all incoming data sets are managed in an Esri cloud-based GIS built on a Microsoft Azure platform, which can manage mobile applications, inputs in real-time from an IoT network, imagery, maps, and KGs that deploy the capabilities mentioned in the previous sections.

Representing USDA's effort to make knowledge available to all via the NAL, the KG GIS capability is also being organized for interoperability through use of the USDA NALT, introduced above. Specifically, the NALT is being included to promote search and discovery across USDA programs (Find and Ask the Expert), as previously suggested [47] in the definition of collaborative KM. The NALT is modeled on the SKOS model and based on GACS. It is true to the first letter in SKOS, which stands for 'simple', in that it provides minimal semantics towards the goal of interoperability. This significantly eases the burden of curation of controlled vocabularies, which is critical as the demands for data interoperability continue to increase while the funding and staffing available for detailed creation and curation of metadata standards is often limited. The ability to add multiple sub-schemes as a concept space will allow the NALT to serve as a standard controlled agricultural vocabulary, with simultaneous domain specificity when needed. Further, the SKOS model does not require users to agree on a single nomenclature for entities—beyond selection of a preferred label, and unlimited synonyms possible to enhance machine discoverability. This feature alone, when fully applied could significantly enhance data interoperability coming from different data types, streams, and owners. This is a flexibility uncommon in most approaches to developing standardized metadata. The NALT currently contains 76,932 preferred label terms, plus 68,823 synonym terms, and 109,074 hidden label terms, and one sub-scheme, NALT Core, with 14,196 preferred label terms, 19,075 synonym terms, and 46,750 hidden label terms.

Partnerships for Data Innovations is also working with the NAL to support development of standard data shapes, utilizing the NALT concepts and iterative communications with research community domains to expand on the 'simple' semantics provided by the NALT, to develop more complex standard data shapes expressions with standard data shape languages being standardized in W3C and the IEEE (IEEE https://standards.ieee.org/ieee/3330/11119/, accessed on 1 February 2023). The NAL is using Shape Expression Language (ShEx) to build standard shapes to validate data as it is incoming or queried for aggregation of properties. Working with researchers to model their data relationships in graph form. These data shapes combined with the NALT URIs will enable data interoperability beyond the constraints of any closed data base or source.

### 4.2.4. Sharing and Reusing Knowledge

By accessing DSI and associated IT infrastructure (Figure 3), and using Protocols.Io, a variety of user CoPs can be served through common protocols, maps, mobile apps, web applications, dashboards, and Hub Sites. Hub Sites manage agency initiatives, to develop CoPs around various large-scale, multi-site, multidisciplinary ARS programs. These initiatives include the Long-Term Agroecosystem Research (LTAR) network, the Greenhouse gas Reduction through Agricultural Carbon Enhancement network (GRACEnet), the Nutrient Use and Outcome Network (NUOnet), Soil Health Assessment network (SHAnet), Agricultural Antibiotic Resistance network (AgAR), Resilient Economic Agricultural Practices network (REAP), and the Dairy Agricultural for People and the Planet network (DAPP). This infrastructure facilitates these communities (CoPs), scattered geographically but with shared interests, coming together virtually or in person to discuss problems and opportunities, best practices, and lessons learned. For example, common protocols used for assessing soil health are being developed and shared by the SHAnet CoP via this infrastructure allowing interoperability and combination of soil health data gathered at many different ARS locations.

While PDI is establishing a KM framework for accelerating the development of new farming practices based on sharing and reusing knowledge assets developed by ARS scientists and collaborators, more traditional knowledge is being disseminated through USDA's Farmers.gov portal. The USDA agencies, through an internal knowledge network, provide knowledge to the farming community through Farmers.gov which covers farm insurance, extension activities, and additional knowledge assets related to running an individual farm.

### 4.2.5. Organizational Governance

As suggested by Drucker [51], adoption of KM requires corporate governance to ensure that knowledge is properly captured and curated. In the scientific realm, the peer reviewed publication process has long been the gold standard in ensuring the proper codification of scientific knowledge. Increasingly, however, scientists are being required to include their data, models, and other support collateral as part of the publication process to foster scientific validation, etc. How that knowledge is being produced has value just like the end results, in other words.

The ARS recognizes the need for an organizational-level approach to management of data and other knowledge assets as well as publication to facilitate sharing, and the need for a diversity of skills to define best practices for managing knowledge and for expanding access through citizen science and technical interoperability. The results of two workshops in 2018 and 2019 "Driving Innovation through Data in Agriculture" brought together agricultural librarians, researchers, data managers, extension agents, experiment station personnel, university administrators, and other individuals with expertise in agricultural data production and management and looked at what management of data and other knowledge assets is currently done, and what are the best practices in agriculture, illustrated with a case study [72].

Agricultural Research Service's senior management is transforming ARS from an organization that relied on scattered knowledge and information management practices of individuals who traditionally have a wide range of skill sets from different technology eras to an organization that employs modern ICT approaches to KM. Transforming data management that relied on practices ranging from paper-based tools (i.e., laboratory notebooks) to siloed data on laptops in spreadsheets, which made sense for a time period that lacked mobile technology for data capture and a common platform for data storage, to an agency that uses robust methods for content management of Big Data coming from drone and satellite images, continuous monitoring networks such as soil moisture sensors and weather stations, as well as point data from sample weights, etc. The resultant ICT infrastructure and KM approach will not only lead to higher quality science, but also science that delivers results faster due to the increased ability of researchers to build on others

work more easily. This will, in turn, lead to a faster delivery of knowledge from the "lab" to the "farm" and from the "farm" back to the "lab". To protect this effort, USDA relies on IT security standards and curation processes to ensure that data and other knowledge assets do not get into the hands of bad actors.

The need for IT security standards is a global problem that goes beyond USDA concerns. Examples of the need for superlative cybersecurity such as the recent invasion of Ukraine abound where adjacent countries can gain advantage by targeting agricultural infrastructure via the information space. For this reason, many IT systems are migrating to cloud service providers where these providers can afford to provide the best security, which is critical to maintain their brand.

*4.3. Ensuring Knowledge Exchange with and within the Farmer Community*

Location-specific knowledge of intended beneficiaries, such as farmers, has often been overlooked despite farmers and other intended beneficiaries being best suited to determine which solutions are most pertinent to their specific needs [37,73]. Farmers have knowledge about local production contexts and practices and are themselves key sources of innovation and adaptations of technology to local conditions as part of their farming process [73–75]. Farmers need to be considered generators of tacit knowledge as well as users of explicit knowledge from academic, business, and governmental institutions. Importantly, integration of farmer participation and knowledge into solution development has been shown to increase farmer adoption of new technologies or solutions [76–78]. Solutions to agricultural problems are only impactful if they are used.

Grass-roots research and development efforts that embed farmers in research programs are being utilized to coalesce farmer and scientific knowledge communities [2,40,75,79,80]. On-farm research, where scientific research occurs in farmer fields, is being used to embed scientific research in farm management. This research occurs at scales meaningful to farmers, acknowledges specific farming realities, and creates value through co-learning and the combination of knowledge pools. On-farm experimentation initiatives involve well over 30,000 farms in more than 30 countries globally [75]. Another approach to coalescing farmer and scientific knowledge has been implemented at the International Maize and Wheat Improvement Center (CIMMYT) in Mexico. Here hubs are developed to build a network of farmers, farm advisors, scientists, research centers, and other actors that collaborate around local solutions to enhance productivity and sustainability of cropping systems. Hub participants implement and adapt best practices resulting from research programs and compare them with conventional practices. In this way, long-term knowledge and methods developed by generations of farmers is integrated with modern scientific methodology and technology [40,42]. In China, scientists have been embedded in villages among farmers to facilitate knowledge exchange between scientific and farmer communities in the Science and Technology Backyard program [80]. By living among farmers, scientists have been able to identify local factors that contributed to yields that were lower than attainable yields (e.g., use of seed varieties not suited for local conditions, improper seed planting density, incorrect tillage depth, improper sowing and harvest dates, improper fertilizer regimen); attainable yield being yield achieved using optimal cropping system management. When these limitations and farmers' concerns were addressed, farmers adopted the recommended management practices and improved yield from 68% of attainable yield to 97% [80].

Farmers use many sources of knowledge (e.g., agriculture extension systems, farm advisors, NGOs, regulatory agencies), but for many, informal participatory farmer networks are key. Informal networks that include farmers lead to learning and innovation as well as adoption and successful implementation of new solutions and technologies [38,73,81,82]. Farmer Field Schools have been implemented where groups of farmers meet regularly to gain knowledge and adopt new farming practices; farming practices that can result in higher yields, increased sustainability, and higher incomes [83]. For example, Farmer Field Schools have been used extensively in implementation of Integrated Pest Management (IPM) approaches for more sustainable control of plant pathogens and pests worldwide [38,73,83].

The IPM approaches substitute agronomic and biological approaches for pesticides, but also require more information and management skills of farmers to implement and manage effectively. Farmer Field Schools and their use to collectively create and deploy knowledge of agroecology, problem solving, skills and their group building and development of social capital for collective decision making are one of the important underpinnings leading to development and spread of IPM. Collective information and knowledge matter greatly for IPM approaches, as coordinated, community-scale decision making by many farmers whose farms together cover large landscapes is necessary for successful outcomes [38]. Farmer Field Schools have been implemented where millions of smallholder farmers participated across Asia, Africa, and Latin America. Other participatory learning frameworks have also been rolled-out in developed countries such as the United Kingdom, Denmark, and the United States [38,73]. To facilitate the acquisition of tacit knowledge from farmers, just as embedding scientists facilitated adoption of new practices, it may be useful to embed knowledge engineers in these informal participatory farmer networks to facilitate knowledge capture using published methodologies [49,63,64].

Going forward, possibilities exist for using Humanistic AI to relay knowledge to and from farmers and other end-users in the agricultural enterprise. Consider Apple's Siri, the first commercially successful, scalable application of many of the KM approaches presented in this paper. When coupled with the more formal approaches, conversational systems like Siri provide an opportunity to automate knowledge exchange to both expedite the dissemination of new knowledge from agricultural science to the farmer and facilitate the capture of structured field data from farmers [84]. New, Ag Tech startups like Dexer (https://www.dexerspeed.com/, accessed on 1 February 2023) will catalyze the adoption of these new conversational systems by being built on a strong KM foundation from supporting agricultural institutions. Regardless, the technology is now mature enough for cost conscious industries like agriculture to make investments in tailoring it to agriculture.

## 5. Conclusions

Food security presents a range of challenges. (1) Countries where food security is currently not a concern, which is the case for countries with developed economies in North America, Europe, and Oceana. In these countries, increasing efforts are directed at decreasing cost of production and associated environmental impacts, which can be substantial, through application of technologies such as precision agriculture and biotechnology and at managing the food chain [2,15,85–87]. (2) Countries where food insecurity is most widespread, such as in sub-Sahara Africa, where the challenge is increasing production of food that is affordable to the local population. In these countries current average crop yields are so low that large relative yield increases can be achieved using knowledge of better management regimens and technology, such as improved seed and mineral fertilizer applications [21,88–91]. (3) Countries with rapidly developing economies, such as China, India, Brazil, Mexico, Indonesia, and Vietnam, that have achieved substantial yield increases but are still experiencing the challenge of greater demand for food due to increasing populations and increasing consumption of animal products. Here new knowledge solutions are needed to increase yield trajectories sustainably to meet growing demands for food as rates of yield increases have slowed despite increased, and sometimes substantial overapplication, of nitrogen and phosphorus fertilizer. This, in turn, leads to environmental pollution in the form of eutrophication, greenhouse gas emissions, and soil acidification [2,24,79,87,89]. Contributing to the challenge in some of these rapidly developing countries is the production of crops on hundreds of millions of small holder farms where certain advanced agricultural technologies are not easily adapted [2].

On top of these regional food security and environmental issues, location-specific solutions will be needed. There are more than 570 million farms worldwide, of vastly different size, occupying almost all the world's climates and soil types, and growing numerous crops using many different crop production systems. In short, due to this tremendous variety in farm size, geography, climate, environmental conditions, and crops

and cropping systems used on the farm, regional solutions will need to be adapted to farm-location-specific needs [10,73,87] and to farmer skill levels and available tools such as internet, computers, and computer literacy [81].

Despite the myriad agricultural solutions required for these location- and farmer-specific problems, the need for KM is constant. The ICT-leveraged KM, when implemented judiciously, has been acknowledged as very useful in facilitating knowledge sharing and access to sources of knowledge [37,46,82] while involvement of farmers and their tacit knowledge leads to adoption. Implementation of ICT-leveraged KM within organizations, like is being done at the USDA, needs to be done globally and these organizational KM networks linked to form a network of KM networks to break down barriers of siloed agricultural knowledge that is scattered globally. Time is at a premium and the goal should be to use knowledge stores located anywhere to develop location-specific solutions in real time globally.

Agriculture has lagged other industry sectors in terms of adoption of ICT. It has been suggested [34] that this lack of adoption is due to " ... [the] lag between invention of new ICT tools and their application, but also by an underinvestment in agriculture research, particularly in non-proprietary public good research ... ". Nevertheless, this lag in adoption puts the cost-conscious Ag industry at risk to the "boom and bust" trends in the IT industry as witnessed by the various AI winters and downturns, or what Gartner implies through its "Hype Cycle" and definition of "technical debt". In other words, the global agricultural enterprise can benefit from both the business KM approach around cost optimization and the academic formalisms without committing the same mistakes from the early days of KM. Regardless, the role of government will continue to be critical for ensuring continuity not only for fostering new innovations and frameworks from academia (i.e., public good research), but also in protecting the agricultural sector from the mercurial nature of the marketplace. In the US, this has certainly been a role that the USDA has played through its partnerships with academia and industry since its inception in the 1860s.

**Author Contributions:** Conceptualization, N.M.S.J., M.J.W.-G., M.D.B. and D.P.R.; writing—original draft preparation, N.M.S.J., M.J.W.-G. and D.P.R.; writing—review and editing, N.M.S.J., M.J.W.-G., M.D.B. and D.P.R. All authors have read and agreed to the published version of the manuscript.

**Funding:** This research received no external funding.

**Institutional Review Board Statement:** Not applicable.

**Informed Consent Statement:** Not applicable.

**Data Availability Statement:** Not applicable.

**Acknowledgments:** The authors acknowledge the Partnerships for Data Innovations (PDI), an initiative of the USDA Agricultural Research Service, for their role in providing information for this manuscript (https://pdi.scinet.usda.gov/, accessed on 1 February 2023).

**Conflicts of Interest:** The authors declare no conflict of interest.

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
