# Peer review of "Scalable Knowledge Management to Meet Global 21st Century Challenges in Agriculture"

_land, doi:10.3390/land12030588_

Round 1

Reviewer 1 Report

Given the impetus to enhance access to knowledge for diverse users, how do the authors propose to bridge both technology and possibly also base-knowledge gaps in how to use this knowledge? The authors refer to Farmer Field Schools and provide an example related to IPM. Are there plans afoot at USDA to really develop this mode of delivery (or others) in order to make this reservoir of knowledge accessible to all?

The authors emphasize the recruitment of tacit knowledge, but such knowledge is typically highly contextualized. How will this context be retained, transformed, and communicated?

Author Response

Responses to Reviewer #1:

Reviewer #1 wanted to know how we plan to bridge both technology and possibly also base-knowledge gaps in how knowledge is used.  We included new text regarding capture and dissemination of tacit knowledge in the revised manuscript (L 142 – 148, L 361 – 364, L 683 – 686). Because tacit knowledge is captured by organizations through methodologies cited in these revised portions of the text, the organizations are ultimately responsible for bridging the gaps.  Inter-organizational communication or knowledge transfer must rely on standards, protocols, and information systems to drive the process of closing these gaps.

Reviewer #1 wanted to know if the USDA has plans to develop modes of delivery of USDA’s reservoir of knowledge to all.  We included text in the revised document detailing how USDA uses the National Agricultural Library and the USDA’s Farmers.gov portal for this purpose (L 534 – 553, L 581 - 587) for dissemination of both scientific and other information related to agriculture.

Reviewer #1 wanted to know how tacit knowledge context will be retained, transformed, and communicated. We included text in the revised document directed at this comment regarding both how USDA is working on this aspect of Knowledge Management as well as the agricultural community in general.  Please see L 142 – 148; L 361 – 364; L 683 – 686).

Reviewer 2 Report

Thank you for writing this very informative paper.  In itself, much institutional, perhaps tacit,  knowledge of the evolution of ICT/KM resources is provided for the benefit of a larger audience.  There are a few suggestions provided in the marked-up version I provided.  In addition, I would like for you to consider making some remarks about how such an envisioned global system could/should be protected from the potentiality of such information being used by nefarious characters.  The recent war in the Ukraine might provide a good example whereby an adjacent country would likely benefit from such information in its targeting of important infrastructure necessary for the Ukrainian ag sector to produce food for her people...and many other developing countries...putting global food security at increased risk.  

Author Response

Responses to Reviewer #2:

Reviewer #2 wanted some remarks about how a Knowledge Management system could/should be protected from nefarious characters.  We provided the following sentence and paragraph in the revised manuscript regarding how organizations make IT networks more secure. 

…….To protect this effort, USDA relies on IT security standards and curation processes to ensure that data and other knowledge assets do not get into the hands of bad actors. 

The need for IT security standards is a global problem that goes beyond USDA concerns.  Examples of the need for superlative cybersecurity such as the recent invasion of Ukraine abound where adjacent countries can gain advantage by targeting agricultural infrastructure via the information space.  For this reason, many IT systems are migrating to cloud service providers where these providers can afford to provide the best security, which is critical to maintain their brand.

Please see L 619 - 627.

Reviewer #2 Comments within the manuscript:

Comment #1. We inserted the word “the” at L133 as requested.

Comment #2. The sentence was rewritten for clarity as requested by Reviewer #2.  Please see L 193 – 194 in the revised manuscript.

Comment #3. Language was added to the revised manuscript to help the Reader understand Figure 1.  Please see L 327 – 330 in the revised manuscript.

Comment #4. A sentence was added to the end of the first paragraph in Section 4.2.4 for impact as request by Reviewer #2.  Please see L 577 – 580 in the revised manuscript.